# Stress-Protective Role of Dietary α-Tocopherol Supplementation in Longfin Yellowtail (*Seriola rivoliana*) Juveniles

Gloria Gertrudys Asencio-Alcudia [1,2,3], Cesar Antonio Sepúlveda-Quiroz [2,3], Juan Carlos Pérez-Urbiola [1], María del Carmen Rodríguez-Jaramillo [1], Andressa Teles [1], Joan Sebastián Salas-Leiva [4], Rafael Martínez-García [2], Luis Daniel Jiménez-Martínez [5], Mario Galaviz [6], Dariel Tovar-Ramírez [1,*,†] and Carlos Alfonso Alvarez-González [2,*,†]

[1] Aquaculture Program, Centro de Investigaciones Biologicas del Noroeste (CIBNOR), Av. Instituto Politecnico Nacional #195, Playa Palo de Santa Rita Sur, La Paz C.P. 23096, Baja California Sur, Mexico; yoya_asencio@live.com.mx (G.G.A.-A.); jperez@cibnor.mx (J.C.P.-U.); cjaramillo63@gmail.com (M.d.C.R.-J.); and.teles84@gmail.com (A.T.)

[2] Laboratorio de Fisiología en Recursos Acuáticos, División Académica de Ciencias Biológicas, Universidad Juárez Autónoma de Tabasco (DACBiol-UJAT), Carretera Villahermosa-Cárdenas Km. 0.5, Entronque Bosques de Saloya, Villahermosa C.P. 86039, Tabasco, Mexico; casq15@gmail.com (C.A.S.-Q.); biologomartinez@hotmail.com (R.M.-G.)

[3] Tecnológico Nacional de México Campus Villahermosa, Km. 3.5 Carretera Villahermosa–Frontera, Cd. Industrial, Villahermosa C.P. 86010, Tabasco, Mexico

[4] Departamento de Medio Ambiente y Energía, CONAHCyT-CIMAV, Miguel de Cervantes No. 120, Complejo Industrial Chihuahua, Chihuhua C.P. 31136, Chih, Mexico; joan.salas@cimav.edu.mx

[5] División Académica Multidisciplinaria de Jalpa de Méndez, Universidad Juárez Autónoma de Tabasco (DAMJM-UJAT), Carretera Estatal Libre Villahermosa-Comalcalco Km. 27+000 s/n Ranchería Ribera Alta, Jalpa de Méndez C.P. 86205, Tabasco, Mexico; luisd1984@hotmail.com

[6] Facultad de Ciencias Marinas, Universidad Autónoma de Baja California, Carretera Transpeninsular Ensenada—Tijuana No. 3917, Col. Playitas, Ensenada C.P. 22860, Baja California, Mexico; mgalaviz@uabc.edu.mx

\* Correspondence: dtovar04@cibnor.mx (D.T.-R.); alvarez_alfonso@hotmail.com (C.A.A.-G.); Tel.: +52-9933581500 (ext. 6480) (C.A.A.-G.)

† These authors contributed equally to this work.

**Abstract:** Aquaculture practices expose fish to several factors that may generate stress, modifying the balance between the production of reactive oxygen species (ROS) and the activity of antioxidant defenses that induce cell damage. Alpha-tocopherol (VE) improves the antioxidant capacity against ROS production in fish. A 50-day trial with longfin yellowtail (*Seriola rivoliana*) juveniles was conducted to compare the dietary supplementation of 500 mg/kg of VE against a control diet without VE supplementation on growth, lymphoid tissue enzymatic activity, immune-system-related gene expression, and the histology of the liver and spleen. Growth, weight gain, specific growth rate, feed conversion rate, and survival did not show significant differences ($p > 0.05$) among treatments. Fish fed with an α-tocopherol-enriched diet showed a higher enzymatic activity of superoxide dismutase in the liver ($p < 0.05$) and a lower percentage of melanomacrophage coverage area in the lymphoid organs ($p < 0.05$). Overexpression was observed of *MyD88* and *il-10* in the spleen, and *il-1b* in the liver in fish fed 500 mg/kg of VE, as well as overexpression of Toll-like 3 in the head kidney, spleen, and liver in fish fed the control diet. Dietary supplementation with VE reduces the effects of oxidative stress and improves lymphoid tissue defense and immune-related gene expression in *S. rivoliana*.

**Keywords:** additives; vitamin E; redox system; antioxidant activity; melanomacrophages

**Key Contribution:** Supplementation with 500 mg/kg of vitamin E in the diet of *S. rivoliana* juveniles overexpresses immune system genes and reduces the number of MMCs in the spleen.

## 1. Introduction

The worldwide exponential demographic growth has pushed scientific and technological progress to search for sustainable and effective options to meet the demand for protein sources without altering organisms' taste, quality, and health [1]. The studies on the Carangidae family have allowed the development of its culture technology worldwide (e.g., in the United States, Japan, and Australia), being considered nowadays as an emerging marine aquaculture species [2]. In this aspect, the longfin yellowtail (*Seriola rivoliana*) presents all these biological and zootechnical characteristics that place this species as a good candidate for aquaculture diversification, including easy handling, rapid growth, and great adaptability to captivity conditions [3]. Furthermore, the actual knowledge available on its biology and larviculture has increased recently [4–7]. Nevertheless, a better dietary formulation to cover this fish species' nutritional requirements might also help toward regular and better production of early juveniles.

Functional feeds are recognized for promoting farmed animals' growth, welfare, and health, coupled with an improvement and/or modulation of their immune system, and providing physiological benefits beyond traditional feeding practices [8]. Among feed additives, vitamins are widely used in farmed freshwater and marine fish, as well as other taxonomic groups of economic importance. They are essential nutrients for the development and growth of vertebrates, but since they are not de novo synthesized by organisms or are synthesized at low rates, they must be included in the diets [9]. Vitamin C and E are major antioxidant additives used in the food industry and have been shown to reduce oxidative stress in animals, avoiding lipid peroxidation when administered in adequate ratios because they can exert synergistic antioxidant effects [10]. Vitamin E is an essential nutrient for fish and plays several vital roles in their health. Vitamin E is a fat-soluble micronutrient that acts as a cell protector from free radical damage due to its antioxidative capacity [11]. Free radicals are oxidative molecules that damage cell membranes, proteins, and DNA [12]. In this sense, vitamin E helps to prevent this damage, avoiding several health problems, including oxidative stress, inflammation, and cancer. In addition, vitamin E has other vital functions in fish nutrition, enhancing red blood cell proliferation and the immune system, reproduction, and the health of the skin and fins [13].

Once vitamin E reaches the digestive system, it is absorbed with other fat-soluble vitamins and transported into the bloodstream by lipoproteins. The main lipoproteins that transport vitamin E throughout the body via the bloodstream are high-density lipoproteins (HDLs) and low-density lipoproteins (LDLs) [14]. Vitamin E is found in all tissues, but it is highest in the liver, lungs, and kidneys, where it is metabolized into several different molecules with different functions; however, the specific metabolic pathways involved in vitamin E utilization vary depending on the form of vitamin E. In this sense, there are four primary forms of vitamin E: α-tocopherol, β-tocopherol, γ-tocopherol, and δ-tocopherol [15]. Alpha-tocopherol is the most active form of vitamin E and is the form that is most found in supplements [16].

In this aspect, the recommended dietary intake of vitamin E for fish varies depending on the species and the life stage; however, most fish generally require a minimum of 200 to 500 milligrams of vitamin E per kilogram of diet, and synthetic or natural sources of vitamin E, such as fish oil, algae, and leafy green vegetables [17]. Specifically, studies on the addition of Vit E in species of the genus *Seriola* are limited. In this regard, Le et al. [18] evaluated the addition of Vit E (40 and 180 mg/kg in feed) and selenium in diets for juvenile *Seriola lalandi* and obtained an increase in blood cell glutathione peroxidase activity. On the contrary, vitamin E deficiency is a relatively rare condition in fish, but it can occur if they are not receiving enough of the nutrients in their diet [19]. Symptoms of vitamin E deficiency include low growth rate, failure of the immune function, and increased susceptibility to disease [20]. Vitamin E has been used in several freshwater or marine species, such as salmon (*Oncorhynchus tshawytscha* and *O. mykiss*) and discus fish (*Symphysodon haraldi*), where appropriate doses increase antioxidant capacity, decreasing total superoxide dismutase (SOD) activity [21] and reducing oxidative stress [22], improving the growth per-

formance and health condition [23]. This study evaluated the effect of VE supplementation on growth, lymphoid tissue enzymatic activity, immune-system-related gene expression, and the histological status of the liver and spleen in *Seriola rivoliana* juveniles.

## 2. Materials and Methods

### 2.1. Rearing, Sampling, and Growth Measurements

Thirty days post-hatch *Seriola rivoliana* juveniles (3.17 ± 0.17 g) were donated by Rancheros del Mar S.A. de C.V. Fish were allowed to acclimatize during 1 week at the Centro de Investigaciones Biológicas del Noroeste S.C. (La Paz, Baja California Sur, Mexico). This nutritional trial consisted of two treatments: a supplemented diet (500 mg/Kg of α-tocopherol) and a control diet (without supplementation of α-tocopherol; Marine Mx, Skretting). Each treatment was performed in triplicates with 20 juveniles per 100 L tank connected to an open water circulation system. Water quality parameters such as dissolved oxygen (6.0 ± 1.0 mg/L), temperature (24 ± 1 °C), and salinity (37 ± 1 ppm) were monitored daily. Fish were maintained under a natural photoperiod (12 h light: 12 h dark).

Longfin yellowtail juveniles were fed for 45 days with a Marine Mx fish Skretting commercial diet (46% crude protein, 12% crude fat, 1% crude fiber, and 12% ash) that contains fish meal, wheat whole, wheat gluten, fish oil, corn gluten, wheat flour squid meal, taurine, and L-Ascorbyl-2 polyphosphate (Vit C) as the primary nutrients, which was finely ground (80 μm) and mixed with water (400 mL/Kg of dry feed), and α-tocopherol (Ref T3376 Sigma-Aldrich, St. Louis, MO, USA) was added to achieve a 500 mg/Kg α-tocopherol dry weight for the supplemented treatment. The feeding rate was established at 5% body weight and adjusted as the fish grew according to biometric monitoring every 15 days. The control diet consisted of the same commercial diet treated in the same manner but without α-tocopherol supplementation. Diets were re-pelleted at a 2 mm diameter and dried at 40 °C for 24 h. The proximate composition of the experimental diets is shown in Table 1. Briefly, the protein was calculated according to the Kjedahl method, and total lipids were calculated according to Folch et al. [24]; the moisture percentage was calculated after 105 °C heating and ash content after incineration (550 °C) was calculated with the weight difference. Both diets (control and 500 mg/Kg of α-tocopherol supplementation) were stored at 4 °C until use. To determine the final concentration of alpha-tocopherol in the control and supplemented with 500 mg/kg diets, samples of both repelletized feeds were taken and sent to the Centro de Investigación en Alimentación y Desarrollo (CIAD) for an analysis with high-performance liquid chromatography on the reversed phase (ÄKTA pure™ 25 M, Cytiva, Marlborough, MA, USA) according to Hess et al. [25]. Briefly, the HPLC method is based on the initial formation of S-carboxy-methyl derivatives of free thiols followed by the conversion of free amino groups to 2,4-dinitrophenyl (DNP) derivatives. Following derivatization, nanomole levels of individual sulfur-containing amino acids are measured using UV detection at 365 nm after separation with reverse-phase ion-exchange HPLC. DNP derivatives of acidic amino acids (including thiol-containing compounds) are separated on a 3-aminopropyl column with reversed-phase ion-exchange HPLC. In the mobile phase, methanol is used to rapidly elute the excess 2,4-dinitrophenol and the DNP derivatives of basic and neutral amino acids. Acetic acid is present in the mobile phase to maintain the bonded-phase amino groups in the protonated form. By increasing the sodium acetate concentration of the mobile phase, selective elution of acidic DNP derivatives is accomplished.

### 2.2. Growth Evaluation and Sampling

Every 15 days, the total length and body wet weight were calculated. For each biometric measurement, fish were previously anesthetized with 8 mL/L of 2-phenoxyethanol (56753 Sigma-Aldrich, St. Louis, MO, USA). After that, daily feed consumption was calculated. With the growth data, the Fulton Condition Index (Fulton's K) was calculated for both treatments. Fish were euthanized with an overdose of 2-phenoxyethanol to collect the

liver, spleen, and head kidney from three fish of each tank ($n$ = 9 per dietary treatment) for histological, biochemical, and gene expression analyses.

**Table 1.** Proximate analysis of commercial feed with and without (control) VE supplementation.

| Treatment | Proteins * % | Lipids % | Humidity % | Ash % | NFE ** % | ∝-Tocopherol ** |
|---|---|---|---|---|---|---|
| Control | 47.85 | 16.01 | 10.1 | 10.38 | 15.66 | 9.95 |
| 500 mg/kg of ∝-tocopherol | 46.72 | 14.83 | 12.66 | 10.29 | 15.5 | 508.35 |

* Protein content was determined using protein factor (6.25). ** NFE—Nitrogen-free extract: % of non-evaluated nutrients. Nitrogen-free extract (NFE) = 1000 − (moisture g $kg^{-1}$ + crude protein g $kg^{-1}$ + ether extract g $kg^{-1}$ + ash g $kg^{-1}$ + crude fiber g $kg^{-1}$).

### 2.3. Histological Analyses

For histological analyses, tissues were fixed in Davidson's solution and subsequently dehydrated in ascending ethanol concentrations of 70% to 100%, and finally individually embedded in paraffin and sectioned in sagittal sections (5 μm), using a rotary microtome (Leica RM 2155, Leica Microsystems, Deerfield, IL, USA). Tissues were stained using hematoxylin-eosin (H&E) for general histological observations and mounted in synthetic resin (Entellan, Merck Millipore, MO, USA). Tissues were examined under a light microscope (Olimpus BX41, Tokyo, Japan) and digitalized at high resolution (600 dpi; 20×). Three randomly selected images from each section were processed with Image-Pro Premier v.9.0 (Media Cybernetics, Silver Spring, MD, USA). The percentage of coverage area for melanomacrophages (MMCs) in the head kidney and spleen was measured. In all cases, the standardized method of an image analysis was adopted as follows [26]. Head kidney melanomacrophage corps (MMCs): HK-MMC (%) = MMCA/TIA × 100 and spleen-MMC (%) = MMCA/TIA × 100 (where MMCA is the melanomacrophage area (in $μm^2$), and TIA is the total area of the image ($μm^2$)). HK-MMC and spleen-MMC are the % covered area of melanomacrophages in the kidney and spleen, respectively.

### 2.4. Antioxidant Enzymatic Activities

Tissue segments (spleen and liver) were homogenized using Ultraturrax (Wilmington, IKA, NC, USA) in 5 volumes ($w/v$) of cold distilled water (4 °C) and centrifuged at 16,000× $g$ for 15 min at 4 °C. The supernatant was retrieved and placed into an ultrafreeze at −80 °C until use. Liver and spleen glutathione peroxidase (GPx) levels were determined kinetically using the Glutathione Peroxidase Cellular Activity Assay kit (Sigma-Aldrich, MI, USA). Spleen and liver catalase (CAT) enzyme activities were determined using a specific kinetic kit (Cayman Chem., Ann Arbor, MI, USA). The superoxide dismutase (SOD) assay kit (Sigma-Aldrich Corp., MI, USA) was used to examine the spleen and liver activities of SOD. Measurements of antioxidant activities were performed on a Microplate Absorbance Spectrophotometer XMark (Biorad, Hercules, CA, USA). CAT and SOD activities were measured at the endpoint after adding the enzyme extract of each tissue with a 20 min incubation at 540 and 450 nm, respectively, and GPx activity was performed using kinetics with absorbance value measurements (340 nm) every 10 sec for 1 min. Antioxidant enzyme activities were normalized with protein content using the Bradford technique [27]. Activities for each SOD, GPx (U/mg of protein), and CAT (nm/min/mg of protein) were calculated according to the supplier's instructions.

### 2.5. Immune System Gene Expression Analysis

For the evaluation of interleukin-10 receptor subunit beta (*il-10*), interleukin-1β (*il-1b*), Toll-like receptor 3 (*Toll-like 3*), and myeloid differentiation primary response 88 (*MyD88*), the spleen and liver for three fish per replicate (n = 9 per treatment) were preserved in RNA later (volume, 1:10; volume/weight) for 24 h and then stored at ~80 °C until use. Total RNA was extracted with a trizol reagent (Invitrogen® reagent code) according to the manufacturer's instructions and total RNA was quantified spectrophotometrically

(A260/280) using a nanodrop Jenway®. Subsequently, cDNA synthesis was performed from 1.5 µg of total RNA from the samples, under the specifications of the iScriptTM Reverse Transcription Supermix for an RT-PCR kit (BIO-RAD®, Hercules, CA, USA), using oligo dT and/or primers in a final reaction of 20 µL per sample. The cDNA was stored at ~80 °C until a further analysis. Relative gene expression was performed with a qPCR analysis, using a CFX96 Touch™ Real-Time Thermal Cycler CFX96 (Bio-Rad) in a total volume reaction of 10 µL per sample, where 5 µL of SsoAdvancedTM Universal SYBR® Green Supermix (Bio-Rad), 4.5 µL of cDNA, and 0.5 µL of primers (10 µmol) were used. All reactions were performed using the following conditions: 1 cycle of 50 °C for 2 min, followed by 1 cycle of 95 °C for 10 min, 40 cycles of 95 °C for 15 s, and 60 °C for 1 min. They ended with a melting curve under standard 60-cycle program conditions to confirm amplification of a single product in each reaction. Primers for this study were designed from the transcriptome of *S. rivoliana* (BioSample accessions SAMN20923996, SAMN20923997, SAMN20923998, SAMN20923999, SAMN20924000, SAMN20924001, SAMN20924002, SAMN20924003), and *18s* (β-actin) and *ef1α* (elongation factor 1α) were used as reference genes (Table 2), where the relative expression was calculated from the average of both reference genes compared to the control [28] according to the best algorithm fit from the RefFinder tool, https://blooge.cn/RefFinder/ (accessed on 15 January 2023).

**Table 2.** Oligonucleotide reference sequence for immune system genes for qPCR of *S. rivoliana*.

| Gene Name | Symbol | Primer Sequence (5′-3′) | Amplicon Size (pb) |
|---|---|---|---|
| β-actin | *18s* | Fw 5′CTGAACTGGGGGCCATGATTAAGAG<br>Rev 5′GGTATCTGATCGTCGTCGAACCTC | 165 |
| Elongation factor 1 | *ef1α* | Fw 5′ TGGTGTTGGTGAGTTTGAGG 3′<br>Rev 5′ CGCTCACTTCCTTGGTGATT 3′ | 173 |
| Interleukin-10 receptor subunit beta | *il-10* | Fw 5′ACAGTGGTATCAGGGATCCTCA<br>Rev 5′CCGACTGTGTAGGGTATGACTG | 155 |
| Interleukin-1β | *il-1b* | Fw 5′AGCCAGCAGAGACACTTAG<br>Rev 5′TGGGTAAAGGTGGCAAGTAG | 124 |
| Toll-like receptor 3 | *Toll-like 3* | Fw 5′CAAATGTTACCAGATTGCCAAACC 3′<br>Rev 5′ TTACCATCAGCATCGGGACAAC 3′ | 168 |
| Myeloid differentiation primary response 88 | *myd88* | Fw 5′ATGAAGCGACGAAAAACCCC 3′<br>Rev 5′AAGACTGAAGATCCTCCACAATGTC 3′ | 135 |

*2.6. Statistical Analysis*

The normal distribution (Kolmogorov–Smirnov test) and homoscedasticity (Shapiro–Wilk test) for weight, percentage of melanomacrophage coverage in the head kidney and spleen, and redox system enzymatic activities data were verified. A Student *t*-test was applied to compare among treatments for those variables. The non-parametric U Mann–Whitney test was applied to the immune system gene expression. The significance value of 0.05 was used for all statistical tests. Software Graph Pad Prism 10 (La Jolla, CA, USA) was used for all analyses.

## 3. Results

*3.1. Growth Evaluation*

Regarding weight growth (Wf), weight gain (WG), specific growth rate (SGR), feed conversion factor (FCR), and survival, no significant differences ($p > 0.05$) were found between treatments (control and supplemented with 500 mg/kg of VE) (Table 3).

**Table 3.** Initial and final growth, weight gain (WG), specific growth rate (SGR), feed conversion rate (FCR), and survival of *S. rivoliana* fed control and VE supplementation. The values are expressed as mean ± standard deviation (Student *t*-test; $p > 0.05$; $n = 3$).

| Treatment | $W_0$ (g) | $W_f$ (g) | WG (g) | SGR | FCR | Survival (%) |
|---|---|---|---|---|---|---|
| Control | 3.17 ± 0.53 | 70.80 ± 9.48 | 68.50 ± 1.72 | 2.71 ± 0.02 | 0.72 ± 0.15 | 100 |
| 500 mg/kg of ∝-tocopherol | 3.06 ± 0.45 | 68.81 ± 8.81 | 64.47 ± 1.39 | 2.66 ± 0.02 | 1.36 ± 0.17 | 100 |

### *3.2. Histological Analyses*

Histological observation of the tissue sections confirmed that the spleen was surrounded by a capsule with a simple cubic epithelium and thin connective tissue, from which irregular trabeculae emerged and extended into the parenchyma. In *S. rivoliana*, the splenic parenchyma was formed from a network of reticular cells fed by vascular sinusoids. The MMCs of organisms fed with both control (Figure 1a) and VE (Figure 1b) diets were observed to be supported with collagen surrounded by ellipsoids, generally rounded structures with a capillary center and lined with a simple cuboidal epithelium. The kidney of *S. rivoliana* is a mixed organ comprising hematopoietic, reticuloendothelial, endocrine, and excretory elements, which have the basic cellular architecture like another teleost. The MMCs are histologically distinguishable within the tissue for presenting macrophages with distinct pigments such as melanin, hemosiderin, and lipofuscins (Figure 1c,d).

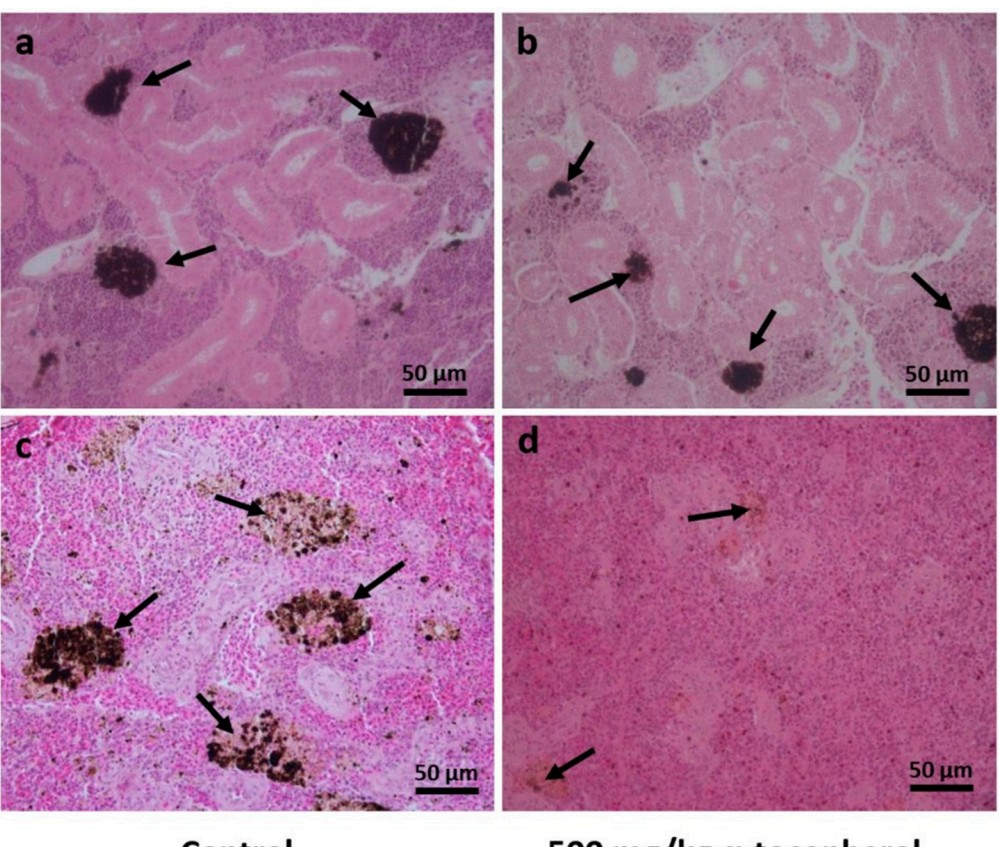

**Figure 1.** Melanomacrophage (MMC) characteristics in *Seriola rivoliana*. H&E stain of (**a**) Head kidney in fish fed control diet, (**b**) Head kidney in fish fed supplemented diet (500 mg/Kg of α-tocopherol), (**c**) Spleen in fish fed control diet, and (**d**) Spleen in fish fed supplemented diet (500 mg/Kg of α-tocopherol). A 40× magnification with arrows indicating MMCs.

The percentage of coverage for MMCs in *S. rivoliana* juveniles' head kidney was statistically higher ($p < 0.05$) in fish fed the control diet ($4.1 \pm 1.9\%$) compared with the fish fed with the VE supplemented diet ($1.9 \pm 0.7\%$) (Figure 2A). Similarly, the MMC coverage area in the spleen for fish fed the control diet ($2.8 \pm 0.4\%$) was statistically higher ($p < 0.05$) when compared with fish fed the α-tocopherol-supplemented diet ($1.2 \pm 0.5\%$) (Figure 2B).

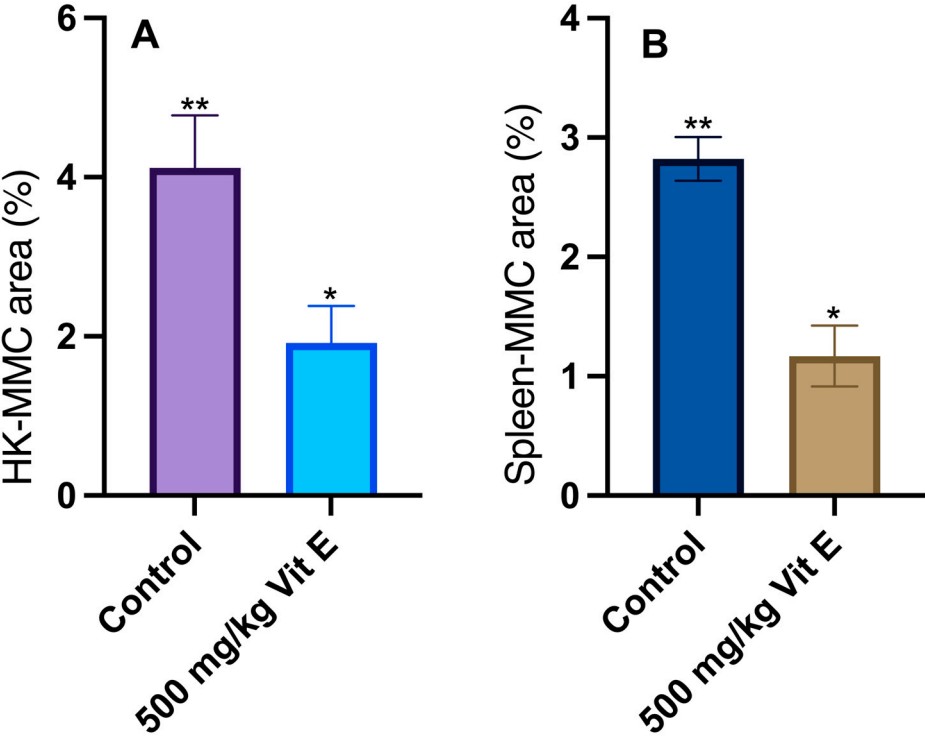

**Figure 2.** Percentage of (**A**) head kidney (HK-MMC) and (**B**) spleen (spleen-MMC) melanomacrophage coverage area in the control and VE-supplemented (with 500 mg/kg of α-tocopherol) groups. The values are expressed as mean ± standard deviation (Student *t*-test; $p < 0.05$; $n = 3$). Asterisks show statistical differences between treatments (Student *t*-test; $p < 0.05$; $n = 3$).

### 3.3. Antioxidant Enzymatic Activities

Significant differences were detected for the SOD activity in the liver (Figure 3A), where fish fed the diet supplemented with vitamin E had higher enzyme activity than fish fed the control diet. However, no significant differences were found for SOD in the spleen. Similarly, statistical differences were not detected ($p > 0.05$) for catalase (Cat) and glutathione peroxidase (GPx) activities between treatments in any of the tissues (Figure 3).

### 3.4. Immune System Gene Expression

Overexpression of the *myd88* gene ($p < 0.05$) was detected in the spleen of fish fed 500 mg/kg of VE compared to those fed the control diet. The head kidney and liver showed no significant differences ($p > 0.05$) for the *myd88* gene between both treatments (Figure 4A). The *il-1b* gene (Figure 4B) showed overexpression in the liver for fish supplemented with VE compared to those fed the control diet, while the head kidney and spleen showed no significant differences ($p > 0.05$) between treatments. The *il-10* gene was overexpressed ($p < 0.05$) in the spleen of fish fed VE compared to fish fed the control diet (Figure 4C), while for the head kidney and liver, no significant differences ($p > 0.05$) were observed between treatments. Finally, overexpression levels for the *Toll-like 3* gene were detected in all tissues ($p < 0.05$) for fish fed the control diet compared to fish fed the VE-supplemented diet (Figure 4D).

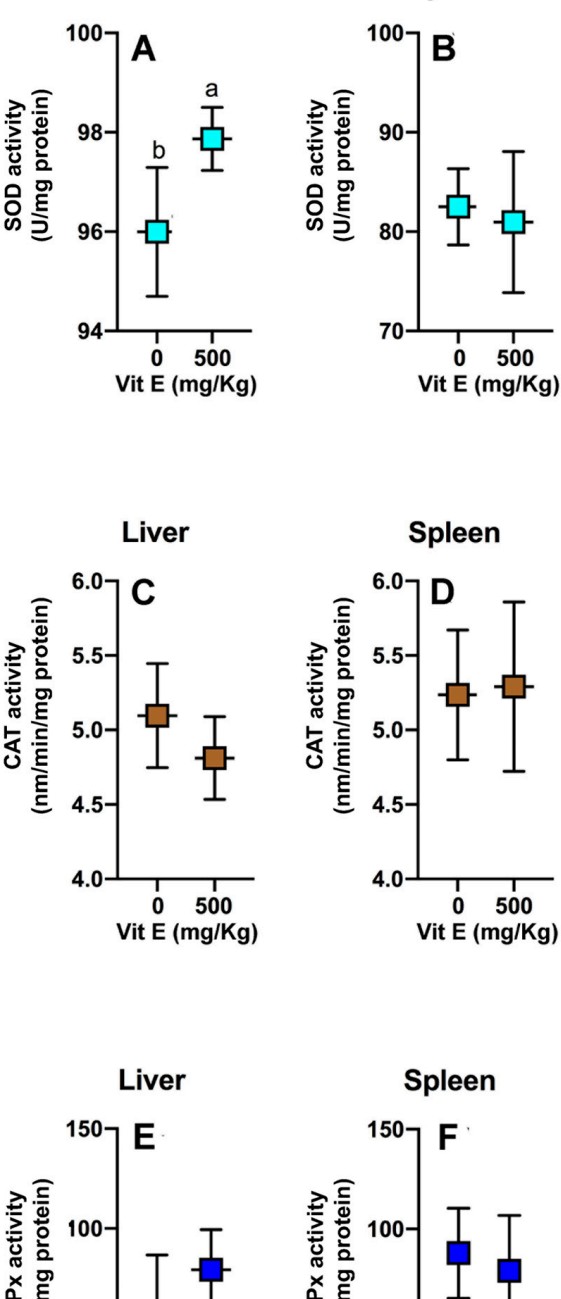

**Figure 3.** Enzymatic activities of the redox system. SOD: Superoxide dismutase activity (U mg protein$^{-1}$) for (**A**) liver and (**B**) spleen, GPx: Glutathione peroxidase activity (U mg protein$^{-1}$) for (**C**) liver and (**D**) spleen, and CAT: Catalase activity (nmol mg protein$^{-1}$) for (**E**) liver and (**F**) spleen. The values are expressed as median ± interquartile range. a,b show statistical differences between treatments (Student *t*-test; $p < 0.05$; $n = 3$).

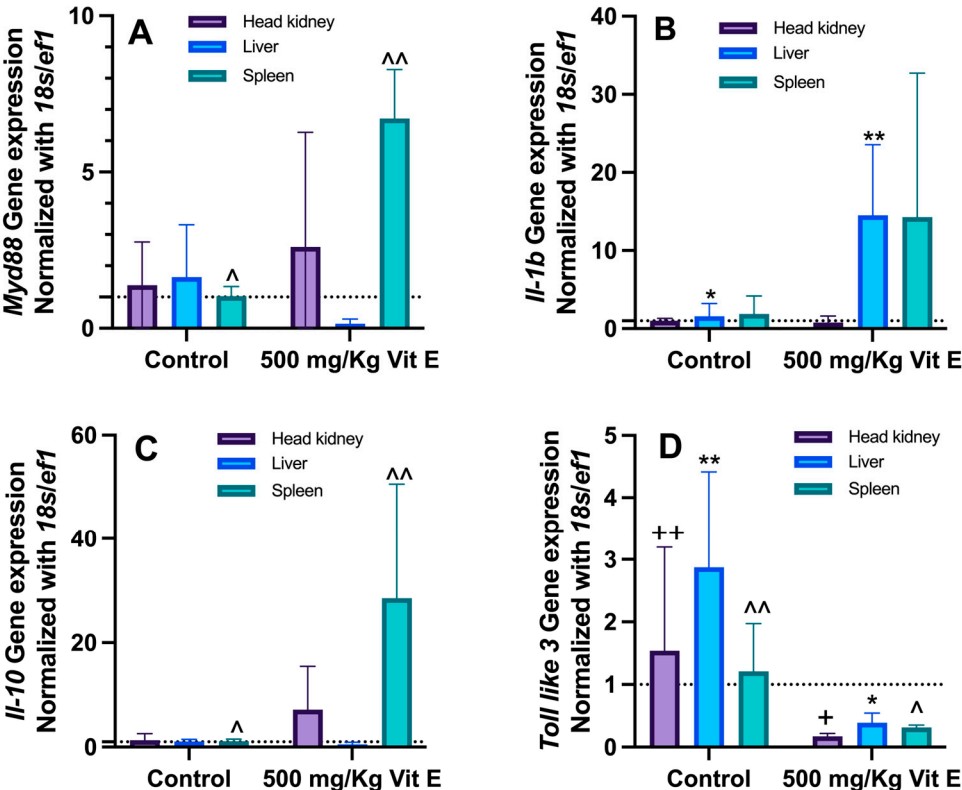

**Figure 4.** Relative gene expression (mean ± SD) of (**A**) Myeloid differentiation primary response 88 (*MyD88*), (**B**) Interleukin-1β (*il-1β*), (**C**) Interleukin-10 receptor subunit beta (*il-10*), and (**D**) Toll factor receptor 3 (*Toll-like 3*) in *S. rivoliana* juveniles. Symbols compare gene expression between treatments (control vs 500 mg/kg Vit E) for head kidney (+), liver (*) and spleen (^). Double symbol indicates differences between treatments. (U Mann–Whitney test; $p < 0.05$; $n = 3$).

## 4. Discussion

According to our results, vitamin E supplementation in *S. rivoliana* juveniles did not show significant differences in growth, food consumption, and survival, which is explained with the fact that this vitamin is not considered a growth promoter and has a different role in organism physiology and metabolism [29]. For that, fish fed 500 mg/Kg of VE increased the SOD activity in the liver, which is one of the target organs that the release of ROS can damage. In this aspect, vitamin E plays an essential role as an antioxidant and in preventing tissue damage because it reduces the accumulation of superoxide radicals and protects cells by donating hydrogen from the phenolic group on the chromanol ring, eliminating peroxyl radicals [30], being one of the main complexes between liposoluble vitamins [31]. In the intestine, tocopherols are hydrolyzed together with the fat from the diet and then secreted into chylomicrons and triacylglycerol, cholesterol, and phospholipids [32–34]. Chylomicron-bound forms of vitamin E are transported through the lymphatic system to peripheral tissue, carrying them through the lipoprotein receptor–mediator process [35] and the remnants are taken by the liver to incorporate the α-tocopherol into lipoproteins through the bindings to the a-tocopherol transfer protein (α-TTP) together with the ATP-binding cassette transporter [36]. Catabolic metabolism of vitamin E begins with one cycle of CYP4F2/CYP3A4-dependent ω-hydroxylation, followed by five subsequent β-oxidation cycles forming the water-soluble end-product carboxyethyl hydroxy chroman [37].

Several studies on dietary vitamins, particularly VE, have recently proved that supplementation is essential for freshwater and marine fish's maintenance, growth, and physiological functions [17,38]. Antioxidant enzymatic systems represented by SOD, CAT, and glutathione peroxidase (GPx) and glutathione reductase (GR) have been detected in most fish and shellfish species investigated to date [39]. Furthermore, SOD, CAT, and GPx

are important components of the first line of the antioxidant defense system and play an essential role in attenuating the potential toxicity of free radicals generated with natural or induced stressors during the life of organisms [40,41]. In the present study, *S. rivoliana* juveniles fed with the control diet (not supplemented with α-tocopherol) showed lower activities for SOD and GPx, which could be related to anti-superoxide anion (ASA) and anti-hydroxyl radical (AHR) capacities, suggesting that antioxidative capacity (elimination of $O_2$) could be affected by α-tocopherol deficiency or a low dose, compared with that from the liver of fish fed the experimental diet (a hypothesis that should be proved in the future). Similar results were obtained in grass carp (*Ctenopharyngodon idella*), where there was a depression of antioxidant enzyme activities, which has been reported to occur due to decreased mRNA levels of the immune organs in fish challenged with *Aeromonas hydrophila* [19]. Additionally, the higher activity of GPx has been related to the inhibition of the antioxidant system with the presence of pathogens or stress [42] where hydrogen peroxide could be involved and partially removed by enzymes such as CAT and GPx [43].

On the other hand, VE supplementation in *S. rivoliana* juveniles decreased the percentage of HK-MMC and spleen-MMC, indicating a beneficial effect on the functionality of these organs compared to fish fed the control diet, which could indicate a VE deficiency. This implies that VE deficiency could induce oxidative damage in many tissues, such as the fish head kidney and spleen. Additionally, other studies associate these damages with the decrease in the anti-superoxide anion and anti-hydroxyl radical capacity, causing oxidative damage perhaps due to the reduction in the radical scavenging capacity in fish [19]. The increment of macrophage aggregates is associated with chronic inflammatory lesions in the tissues [44], which can be diminished when cell membranes and organelles are protected with antioxidant compounds such as VE. MMCs have been used in fish as a biomarker of environmental stress and are an essential component of humoral immune response [45]. Melanomacrophage centers (MMCs) are macrophage-like cells containing different pigments, including lipofuscin, melanin, and hemosiderin, that have been deeply analyzed in Butterfly Splitfin (*Ameca splendens*) [46]; however, the observation of these molecules and aggregates in our experiment was not conducted. They are aggregates of these cells observed in a higher teleost, and these are found randomly and irregularly distributed in organs, mainly hematopoietic, such as the head kidney, spleen, and sometimes in the liver [44], and their concentration increases with age [47], health status [48], nutritional status, and general stress under culture conditions [49,50], as well as tissue damages [51] and water pollution [52].

Regardless that the VE requirements of fish vary with cultured species and size, dietary lipid sources, and culture conditions, in our study, it is evident that dietary supplementation with 500 mg/kg of α-tocopherol in *S. rivoliana* juveniles increases immunocompetence, which is reflected by overexpression of *MyD88* and *il-10* in the spleen and *il-1β* in the liver, as well as overexpression of *Toll-like 3* in fish fed the control diet in all tissues. This overexpression of immune system genes is related to the ability of fish to counteract the effects of various stressors that cause membrane damage with the action of ROS, as well as possible parasitic or bacterial infections during culturing by decreasing their immune response [45]. In this regard, interleukin-1 beta (IL-1b) is a potent pro-inflammatory cytokine that stimulates the production of other inflammatory cytokines and chemokines and increases the expression of adhesion molecules on cells. Il-1b helps recruit immune cells to the site of infection or injury and activates various immune cells, including macrophages, neutrophils, and T cells [53]. In parallel, interleukin 10 (IL-10) is a potent immunosuppressive cytokine that inhibits the production of pro-inflammatory cytokines and chemokines, and it also downregulates the expression of the major histocompatibility complex (MHC) on cells, as well as promotes the development of regulatory T cells (Tregs). To prevent autoimmune diseases and allergic reactions, Tregs play an essential role; finally, IL-1b improves fibroblast proliferation and the collagen production, which helps close wounds [54]. With reference to the myeloid differentiation primary response protein 88 (MyD88), it is an adaptor protein that plays a central role in the Toll-like receptor (TLR) signaling pathway. TLRs are a family

of pattern recognition receptors involved in the innate immune response to infection and injury. In this aspect, *MyD88* is also involved in the signaling pathway for the interleukin-1 receptor (IL-1R) and Toll-like 3. Together, TLRs and IL-1Rs are expressed in various cells in fish, including macrophages, neutrophils, and dendritic cells.

When a TLR or IL-1R is activated, it recruits MyD88 to the plasma membrane, activating a cascade of signaling events that leads to pro-inflammatory cytokines and chemokines [55,56]. Finally, Toll-like receptor 3 (TLR3) is a pattern recognition receptor involved in fish's innate immune response. It is expressed in various cells, including macrophages, neutrophils, and dendritic cells. TLR3 recognizes double-stranded RNA (dsRNA) as a viral replication intermediate. When TLR3 is activated, it recruits adaptor proteins to the plasma membrane. These adaptor proteins then activate a cascade of signaling events that produce pro-inflammatory cytokines and chemokines that recruit immune cells to the site of infection and activate them to kill pathogens and repair damaged tissue [57,58]. Based on the above, there is a direct relationship between antioxidant capacity and the expression of immune system genes, which activate protection against ROS and stimulate the production of defense cells versus possible infections, so the supplementation of 500 mg/kg of α-tocopherol in *S. rivoliana* juveniles promotes these capacities; however, a study with different doses is required to determine the optimal level in the diet of this species.

It is worth adding that VE can exert different effects on growth, health, antioxidant status, and survival according to the time and administered dose, size, and fish species [17]. It has been reported that requirements in VE in marine fish may vary from the highest values of 2900 mg/kg of feed in gilthead seabream (*S. aurata*) to medium values of 800 mg/kg of feed for meagre (*A. regius*) and the lowest of 78–111 mg/kg of feed in Cobia (*R. canadum*). The observed effect varies from increasing larval growth, granulomatosis disappearance, liver VE reserves, antioxidant status, to non-specific immunity [59–61]. Thus, although further dose–response trials should be conducted to determine the optimal dietary VE content in *S. rivoliana* juveniles, present results indicate that a medium supplementation (500 mg/kg of α-tocopherol) of VE in this species is adequate.

## 5. Conclusions

In conclusion, it is paramount to continue the evaluation of the α-tocopherol requirement of *S. rivoliana* to achieve optimal growth and health status during any growth phases. This study indicated that the supplementation of 500 mg/Kg of α-tocopherol improved the antioxidant enzyme activities, modified the coverage area of MMCs in the head kidney and spleen, and improved immune system gene expression in our species. However, it is necessary to conduct a challenge study (crowding, air exposure, high stocking density, and bacterial challenges, among others) to understand the physiological role of α-tocopherol supplementation in metabolic pathways or stressful culture conditions.

**Author Contributions:** Conceptualization, D.T.-R., G.G.A.-A. and C.A.A.-G.; methodology, G.G.A.-A., J.C.P.-U., C.A.S.-Q. and M.d.C.R.-J.; formal analysis, G.G.A.-A., R.M.-G. and L.D.J.-M.; investigation, G.G.A.-A., D.T.-R. and C.A.A.-G.; resources, D.T.-R. and C.A.A.-G.; writing—original draft preparation, G.G.A.-A., D.T.-R., C.A.A.-G., J.S.S.-L., A.T. and M.G.; writing—review and editing, G.G.A.-A., R.M.-G., D.T.-R. and C.A.A.-G.; visualization, G.G.A.-A., R.M.-G. and C.A.A.-G.; supervision, D.T.-R. and C.A.A.-G.; project administration, D.T.-R. and C.A.A.-G.; funding acquisition, D.T.-R. and C.A.A.-G. All authors have read and agreed to the published version of the manuscript.

**Funding:** This research was partially funded by the Consejo Nacional de Humanidades, Ciencias y Tecnología (CONAHCyT) in Mexico, grant number: CB-2016-01-282765. This study was carried out with the collaboration of Red CYTED LARVAplus (117RT0521).

**Institutional Review Board Statement:** The experiment on fishes did not need specific ethical approval in Mexico.

**Informed Consent Statement:** Not applicable.

**Data Availability Statement:** The data that support the findings of this study are available upon request from the authors.

**Acknowledgments:** The authors thank Rancheros del Mar S.C. for providing juveniles, and María Eulalia Meza Chavez for technical assistance in preparing histological sections. The CIBNOR S.C. technical support: Martha Reyes, Patricia Hinojosa, Marcos Quiñones, Ernesto Goytortúa, Pablo Ormart, and Delfino Barajas (during the nutritional trial). Funding was provided by Consejo Nacional de Humanidades, Ciencias y Tecnología (CONAHCYT grant 282765). G.G.A.-A. is grateful to the Consejo Nacional de Humanidades, Ciencias y Tecnología (CONAHCyT, Mexico) for the PhD scholarship (register number: 571169). Collaboration between Ibero-American researchers was carried out under the framework of the network LarvaPlus, "Strategies for the development and improvement of fish larvae production in Ibero-America" (117RT0521), funded by the Ibero-American Program of Science and Technology for Development (CYTED, Spain).

**Conflicts of Interest:** The authors declare no conflict of interest.

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
