# Peer review of "Stress-Protective Role of Dietary α-Tocopherol Supplementation in Longfin Yellowtail (Seriola rivoliana) Juveniles"

_fishes, doi:10.3390/fishes8100526_

Round 1
Reviewer 1 Report
The manuscript entitled "Stress-protective role of dietary ⍺-tocopherol supplementation in the longfin yellowtail (Seriola rivoliana) juveniles" is very well written. The authors present the obtained results in a clear manner, using good quality Figures, which they compare with the results of other authors in the discussion.
To make the article well received and cited, I suggest making some improvements to the main text of manuscript:
line 146 - please specify the information of the manufacturer (microtome and microscope)
lines 150-152 - Were there no MMCs in the liver?
lines 158-169 - How did you interpret the results of the enzymatic analyses? Was a spectrophotometer used? Please complete the description.
lines 212-223 - Was hemosiderin and lipofuscin deposition in MMC checked? I think it would be worthwhile to perform additional staining to confirm or exclude the deposition of these substances (compare with the description in the Discussion, lines 318-320). Please see in Latoszek et al. (2019) - Latoszek, E., Kamaszewski, M., Milczarek, K., Puppel, K., Szudrowicz, H., Adamski, A., ... & Ostaszewska, T. (2019). Histochemical characteristics of macrophages of butterfly splitfin Ameca splendens. Folia Biologica (Kraków), 67(1), 53-60.
Author Response
The manuscript entitled "Stress-protective role of dietary ⍺-tocopherol supplementation in the longfin yellowtail (Seriola rivoliana) juveniles" is very well written. The authors present the obtained results in a clear manner, using good quality Figures, which they compare with the results of other authors in the discussion.
Response: Thanks for your support in improving our work.
To make the article well received and cited, I suggest making some improvements to the main text of the manuscript:
line 146 - please specify the information of the manufacturer (microtome and microscope)
Response: Information was included in the MS.
lines 150-152 - Were there no MMCs in the liver?
Response: We did not detect MMC differences when reviewing the liver samples between the two treatments, so this information is not included in the article. Thank you for your comment.
lines 158-169 - How did you interpret the results of the enzymatic analyses? Was a spectrophotometer used? Please complete the description.
Response: Thanks for your comment, this information was included in the MS (Lines 171-175).
lines 212-223 - Was hemosiderin and lipofuscin deposition in MMC checked? I think it would be worthwhile to perform additional staining to confirm or exclude the deposition of these substances (compare with the description in the Discussion, lines 318-320). Please see in Latoszek et al. (2019) - Latoszek, E., Kamaszewski, M., Milczarek, K., Puppel, K., Szudrowicz, H., Adamski, A., ... & Ostaszewska, T. (2019). Histochemical characteristics of macrophages of butterfly splitfin Ameca splendens. Folia Biologica (Kraków), 67(1), 53-60.
Response: We thank the reviewer for the suggestion for the detection of lipofuscins and hemosiderin in the tissues of our fish; however, now we do not have the Pearls' Prussian blue method technique set up, which does not allow us to make these observations, but we will consider them for the following studies. We included the suggested reference in our MS.
Reviewer 2 Report
Line106: Why was the level of α-tocopherol chosen to be 500mg/kg? What does the literature support?
Line112-116: Why was a commercial feed used? Can you indicate what specific feed ingredients are in the feed? Also, the authors mention in Line 68 that vitamin C and vitamin E have synergistic antioxidant effects, so what is the level of vitamin C in the feed? Are the results of this study affected by vitamin C? Please explain.
Line 130-133: "*** Determined by HPLC-RP at CIAD" and "∝-tocopherol**" are inconsistent. Please explain.
Line154: "MCMA" and "MMCA" are correct? Please check.
Line199-201: Why are different statistical analysis methods chosen for different indicators?
Author Response
Line106: Why was the level of α-tocopherol chosen to be 500mg/kg? What does the literature support?
Response: The adequate amount of vitamin E for the different fish species is highly variable, so 200 to 500 mg/kg of feed is generally included as a supplement in diets (lines 89-90, reference 17). For Seriola species, there was only one precedent on the inclusion of vitamin E and selenium in juvenile Seriola lalandi; however, these authors only evaluated low concentrations of Vit E (40 and 180 mg/kg of feed), so for our experiment, it was decided to include a higher concentration to understand the effects of this micronutrient on the immune system and oxidative stress. This reference was included in the introduction section (Lines 91-92). We thank your comments.
Line112-116: Why was a commercial feed used? Can you indicate what specific feed ingredients are in the feed? Also, the authors mention in Line 68 that vitamin C and vitamin E have synergistic antioxidant effects, so what is the level of vitamin C in the feed? Are the results of this study affected by vitamin C? Please explain.
Response: We included in the manuscript (lines 118-120) the main ingredients that make up the commercial feed, which includes vitamin C (L-Ascorbyl-2 polyphosphate); however, the company does not indicate the quantities of each ingredient. In this regard, we used this commercial feed because it is widely used for marine species, including S. lalandi, since there is no specific commercial feed for S. rivoliana.
We agree with the reviewer's comments regarding the synergy between vitamin E and C; however, the objective of our research focused specifically on Vitamin E on the immune and oxidative stress parameters in S. rivoliana. We plan to conduct a study in the future regarding the interaction of these vitamins in our species, and we thank the reviewer for the suggestion.
Line 130-133: "*** Determined by HPLC-RP at CIAD" and "∝-tocopherol**" are inconsistent. Please explain.
Response: More detail was included in the MS (lines 135-147). Thanks for you observation.
Line154: "MCMA" and "MMCA" are correct? Please check.
Response: Corrected in the MS, thanks for your observation
Line199-201: Why are different statistical analysis methods chosen for different indicators?
Response: Considering the postulates for parametric analysis (Normality and Homoscedasticity), it is necessary that the data obtained be evaluated, and it is determined if they meet these criteria. As indicated in the Statistical Analysis subsection, the data for weight, percentage of melanomacrophages coverage in the head kidney and spleen, and redox system enzymatic activities did meet these criteria; however, the genomic expression did not meet them, so we chose to analyze them with non-parametric statistics (U Mann-Whitney test). We thank you for your comments.
Reviewer 3 Report
As requested, I have reviewed the manuscript titled “Stress-protective role of dietary ⍺-tocopherol supplementation in the longfin yellowtail (Seriola rivoliana) juveniles”.
The study assessed the effects of supplementing diets for juvenile longfin yellowtail (S. rivoliana) with 500 mg/kg of vitamin E (α-tocopherol) compared to fish fed without supplementation. The evaluation focused on parameters such as weight gain and immune system activities.
The results showed that dietary supplementation vitamin E did not lead to significant differences in the zootechnical performance parameters of the animals. However, it did contribute to improvements in certain immune system parameters, such as higher enzymatic activity of superoxide dismutase in the liver and a lower percentage of melanomacrophage coverage area in the lymphoid organs. These findings suggest that such supplementation may help mitigate the effects of oxidative stress and enhance immune system responses in S. rivoliana, particularly benefiting the immune systems of animals facing stressful situations, which are common in aquaculture practices.
While this study only examined one dosage of supplementation, it is challenging to estimate an ideal dosage for inclusion in rations or determine its potential significance in terms of productive performance. Nevertheless, the results pertaining to immune system enhancements are robust, supported by extensive analyses, and are thoroughly discussed in the manuscript. These findings can serve as a foundation for developing more targeted feeds for this species.
The Introduction section is clear and comprehensive, and the Discussion section is well-developed. Therefore, I recommend accepting the manuscript in its current form.
Thank you for considering my opinion.
Author Response
As requested, I have reviewed the manuscript titled “Stress-protective role of dietary ⍺-tocopherol supplementation in the longfin yellowtail (Seriola rivoliana) juveniles”.
The study assessed the effects of supplementing diets for juvenile longfin yellowtail (S. rivoliana) with 500 mg/kg of vitamin E (α-tocopherol) compared to fish fed without supplementation. The evaluation focused on parameters such as weight gain and immune system activities.
The results showed that dietary supplementation vitamin E did not lead to significant differences in the zootechnical performance parameters of the animals. However, it did contribute to improvements in certain immune system parameters, such as higher enzymatic activity of superoxide dismutase in the liver and a lower percentage of melanomacrophage coverage area in the lymphoid organs. These findings suggest that such supplementation may help mitigate the effects of oxidative stress and enhance immune system responses in S. rivoliana, particularly benefiting the immune systems of animals facing stressful situations, which are common in aquaculture practices.
Response: We thanks for the positive comment.
While this study only examined one dosage of supplementation, it is challenging to estimate an ideal dosage for inclusion in rations or determine its potential significance in terms of productive performance. Nevertheless, the results pertaining to immune system enhancements are robust, supported by extensive analyses, and are thoroughly discussed in the manuscript. These findings can serve as a foundation for developing more targeted feeds for this species.
Response: We thanks for the positive comment.
The Introduction section is clear and comprehensive, and the Discussion section is well-developed. Therefore, I recommend accepting the manuscript in its current form.
Response: We thanks for the positive comment.
Thank you for considering my opinion.
Response: We thanks for the positive comment.